# Global warming-induced Asian hydrological climate transition across the Miocene–Pliocene boundary

Hong Ao [1,2✉], Eelco J. Rohling [3,4], Ran Zhang [5✉], Andrew P. Roberts [3], Ann E. Holbourn [6], Jean-Baptiste Ladant [7,8], Guillaume Dupont-Nivet[9,10], Wolfgang Kuhnt[6], Peng Zhang [1,2], Feng Wu[1], Mark J. Dekkers [11], Qingsong Liu[12], Zhonghui Liu [13], Yong Xu[14], Christopher J. Poulsen [7], Alexis Licht [15], Qiang Sun [16], John C. H. Chiang[17], Xiaodong Liu [1], Guoxiong Wu[5], Chao Ma[18], Weijian Zhou [1,2], Zhangdong Jin [1,2], Xinxia Li[19], Xinzhou Li[1], Xianzhe Peng[20], Xiaoke Qiang [1] & Zhisheng An [1,2]

Across the Miocene–Pliocene boundary (MPB; 5.3 million years ago, Ma), late Miocene cooling gave way to the early-to-middle Pliocene Warm Period. This transition, across which atmospheric $CO_2$ concentrations increased to levels similar to present, holds potential for deciphering regional climate responses in Asia—currently home to more than half of the world's population— to global climate change. Here we find that $CO_2$-induced MPB warming both increased summer monsoon moisture transport over East Asia, and enhanced aridification over large parts of Central Asia by increasing evaporation, based on integration of our ~1–2-thousand-year (kyr) resolution summer monsoon records from the Chinese Loess Plateau aeolian red clay with existing terrestrial records, land-sea correlations, and climate model simulations. Our results offer palaeoclimate-based support for 'wet-gets-wetter and dry-gets-drier' projections of future regional hydroclimate responses to sustained anthropogenic forcing. Moreover, our high-resolution monsoon records reveal a dynamic response to eccentricity modulation of solar insolation, with predominant 405-kyr and ~100-kyr periodicities between 8.1 and 3.4 Ma.

[1] State Key Laboratory of Loess and Quaternary Geology, Institute of Earth Environment, Chinese Academy of Sciences, Xi'an, China. [2] Open Studio for Oceanic-Continental Climate and Environment Changes, Pilot National Laboratory for Marine Science and Technology (Qingdao), Qingdao, China. [3] Research School of Earth Sciences, Australian National University, Canberra, ACT, Australia. [4] Ocean and Earth Science, University of Southampton, National Oceanography Centre, Southampton, UK. [5] Institute of Atmospheric Physics, Chinese Academy of Sciences, Beijing, China. [6] Institute of Geosciences, Christian-Albrechts-University, Kiel, Germany. [7] Department of Earth and Environmental Sciences, University of Michigan, Ann Arbor, MI, USA. [8] Laboratoire des Sciences du Climat et de l'Environnement, LSCE/IPSL, CEA-CNRS-UVSQ, Université Paris-Saclay, 91191 Gif-sur-Yvette, France. [9] Géosciences Rennes, UMR-CNRS 6118, University Rennes, Rennes, France. [10] Department of Geosciences, Potsdam University, Potsdam, Germany. [11] Paleomagnetic Laboratory 'Fort Hoofddijk', Department of Earth Sciences, Faculty of Geosciences, Utrecht University, Utrecht, The Netherlands. [12] Department of Ocean Science and Engineering, Southern University of Science and Technology, Shenzhen, China. [13] Department of Earth Sciences, University of Hong Kong, Hong Kong, China. [14] Xi'an Center of Geological Survey, China Geological Survey, Xi'an, China. [15] Department of Earth and Space Sciences, University of Washington, Seattle, WA, USA. [16] College of Geology and Environment, Xi'an University of Science and Technology, Xi'an, China. [17] Department of Geography, University of California, Berkeley, CA, USA. [18] State Key Laboratory of Oil and Gas Reservoir Geology and Exploitation, Chengdu Universityof Technology, Chengdu, China. [19] School of Earth Sciences, China University of Geosciences (Wuhan), Wuhan, China. [20] School of Information Management, Nanjing University, Nanjing, China. ✉email: aohong@ieecas.cn; zhangran@mail.iap.ac.cn

The Miocene–Pliocene boundary (MPB), which is defined formally at the base of the deep marine Mediterranean Trubi marls, occurred at ~5.3 Ma, five orbital precession (19-kyr and 23-kyr) cycles below the C3n.4n palaeomagnetic normal polarity chron[1–4]. In the Mediterranean, the Messinian salinity crisis ended across the MPB[2,3,5]. In a global context, the MPB marks a transition from a late Miocene cooling trend to the early-to-middle Pliocene Warm Period[6,7]. It coincided with an atmospheric $CO_2$ concentration rise of ~100–250 ppm[8–10], an increase with similar amplitude (albeit not the rate) to the current rise due to anthropogenic emissions. The early-to-middle Pliocene Warm Period was the most recent period of persistently warmer-than-present conditions. During this warm interval, the Northern Hemisphere was largely ice-free and atmospheric $CO_2$ concentrations were comparable to present-day levels[8–11]. Detailed analysis of the MPB, therefore, enables investigation of the large-scale climate response to a natural atmospheric $CO_2$ increase and global warming over equilibrium timescales for comparison with rapid out-of-equilibrium present-day responses. Compared to relatively extensive Quaternary and modern climate studies, however, little is known about the detailed terrestrial climate changes related to this global warming event because high-resolution records that offer continuous coverage of distinct orbital variability from the late Miocene to the Pliocene are rare.

Today, Asia has wet monsoonal regions to the south and east of the Tibetan Plateau (~200–2000 mm annual precipitation) and arid continental regions to the north and west of the Tibetan Plateau (<50 to ~100 mm annual precipitation) (Fig. 1a). Arid Central Asia (>35°N) is dominated by upper-tropospheric Westerlies and is essentially beyond the reach of summer monsoon rain (Fig. 1a). Both the Asian monsoon and the Westerlies are critical components of the global atmospheric circulation; they play a key role in the global climate system, impacting surface ocean circulation and driving air-sea heat, moisture, momentum fluxes, and carbon exchange[12–14]. Our present understanding of Asian climate variability and dynamics relies primarily on Quaternary results, when a bipolar icehouse climate state was fully established and global climate was colder than today, with waxing and waning Northern Hemisphere ice sheets on orbital timescales[11]. To better appreciate aspects of Asian climate dynamics and future climate change, it is pertinent to reconstruct sufficiently highly resolved palaeoclimate records to identify orbital timescale variability and to understand the forcing mechanisms of Asian climate change during the warmer-than-present late Miocene–Pliocene.

The Chinese Loess Plateau (CLP, ~640,000 km²) is located at the transition between humid and arid regions in Central China and is sensitive to seasonally alternating southeasterly summer

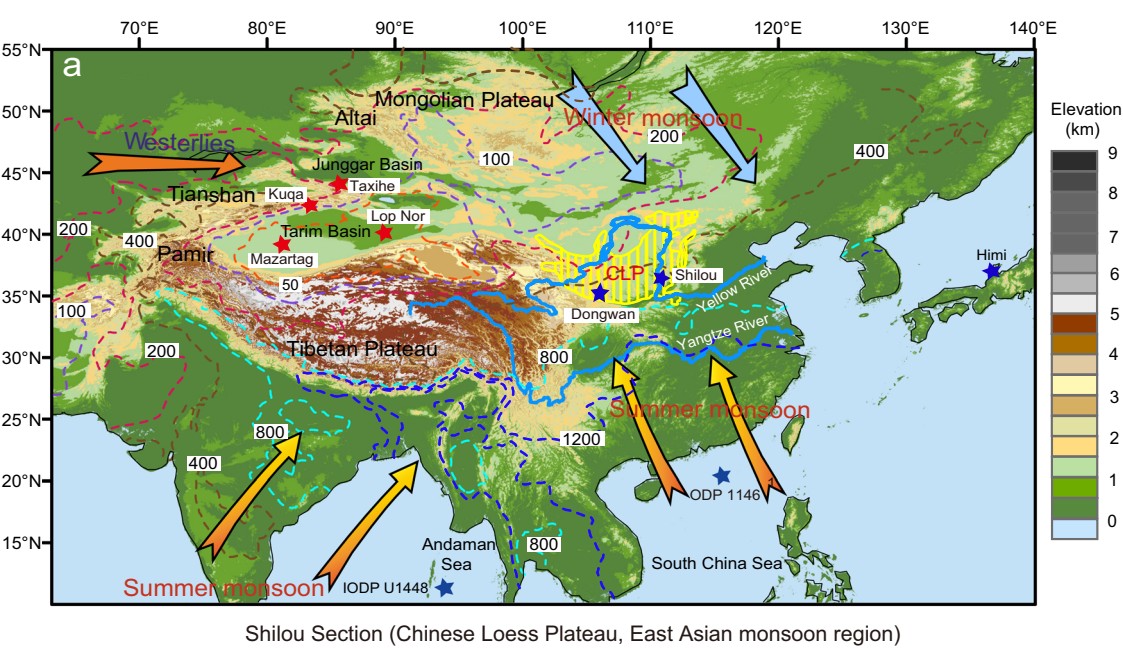

Shilou Section (Chinese Loess Plateau, East Asian monsoon region)

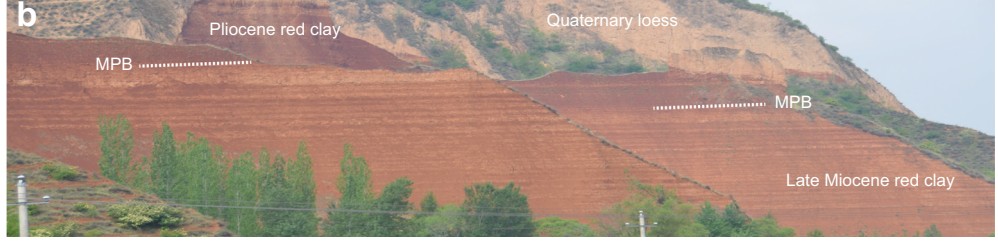

**Fig. 1 Site location map and field photograph. a** Topographic map of Asia with important geographic landmarks mentioned herein. Dominant winds and mean annual precipitation (mm, 1979 to 2007 average) are shown as arrows and dashed contour lines, respectively. Red and blue stars mark locations that shifted to drier and wetter conditions across the Miocene–Pliocene boundary (MPB), respectively. South and East Asia receive high Asian summer monsoon precipitation. Central Asia is a low precipitation region that is influenced mostly by the Westerlies. **b** Field photograph (taken by H. Ao) of the Shilou red clay sequence (with horizontal bedding) from the eastern Chinese Loess Plateau (CLP, monsoonal region). The upper sequence (Pliocene) has a more saturated red colour than the lower part (late Miocene), which is consistent with enhanced pedogenesis and increased summer monsoon precipitation across the MPB.

monsoon and northwesterly winter monsoon circulations[4,15–20] (Fig. 1a). With thicknesses of up to ~600 m, CLP aeolian loess/red clay sequences, which reflect primarily near-surface winter monsoon dust transport from the Central Asian arid regions (i.e., inland Gobi-sandy deserts and wind eroded lands), provide a unique high-resolution archive of terrestrial climate variations that spans continuously from the latest Oligocene to the Quaternary[15–18,21]. At present, orbital climate variability is well constrained in Quaternary loess sequences[22–25], but is poorly resolved in the underlying red clay sequence[18,26] because rapidly measurable magnetic susceptibility (χ), which is used routinely to reveal orbital climate variability of Quaternary loess, does not capture distinct orbital signals in the red clay[19].

Here, we investigate Asian monsoon responses to the MPB global warming event and orbital forcing during the late Miocene–middle Pliocene in a global context by providing ~1–2-kyr resolution ~4.7-Myr-long proxy records (Al/Na, Rb/Sr, and lightness) spanning continuously from ~8.1 to ~3.4 Ma from the Shilou aeolian red clay succession on the eastern CLP (Fig. 1a). This section has an exceptional and distinct cyclostratigraphy that is visible to the naked eye in the field (Fig. 1b) and is, thus, well suited to reveal orbital-scale terrestrial climate signals recorded in the CLP red clay. We integrate our high-resolution records with existing palaeoclimate records from the wider Asian region and land-sea correlations. We also present Earth System model simulation results with late Neogene boundary conditions, to assess the influence of $CO_2$-induced MPB warming on spatial Asian hydroclimate changes. Our comprehensive study can aid assessment of future Asian climate changes as global climate continues to warm.

## Results

The Shilou red clay section (36°55′N, 110°56′E, 1150 m elevation) is located on the northwestern Asian summer monsoon margin, an area sensitive to summer monsoon variability (Fig. 1). It (Figs. 1b and 2a; Supplementary Fig. 1) comprises alternating brownish and light red (carbonate-nodule-rich) late Miocene palaeosols, with dominantly darker red Pliocene palaeosols (see Supplementary Note 1 for lithological details). Al/Na, Rb/Sr, and lightness of Neogene red clay and Quaternary loess-palaeosol sequences across the CLP are used routinely as proxies of regional summer monsoon precipitation[17,26–29] (for a more detailed discussion of proxy validation, see Supplementary Note 1 and Supplementary Fig. 2). We measured these proxies from the Shilou red clay at 2-cm intervals (~1–2-kyr time spacing), which provide the highest-resolution late Miocene–Pliocene CLP red clay palaeoclimate records presently available, to infer prominent monsoon events and variability over orbital time scales and long-term trends. Using the latest magnetochronology for the Shilou red clay section[19], the robustness of which has been addressed in recent studies[4,19,30] (Supplementary Note 2), we developed a combined magneto- and astro-chronology spanning continuously from ca 8.1 to 3.4 Ma by tuning Rb/Sr to Earth's computed orbital eccentricity[31] (see "Methods" and Supplementary Figs. 3–4 for details). In the Shilou red clay section, brownish and dark red palaeosols, which formed under higher precipitation and stronger chemical weathering conditions, have higher Al/Na and Rb/Sr values and lower lightness values than the light red carbonate-nodule-rich palaeosols that formed under lower precipitation and weaker weathering conditions (Fig. 2a–d). These proxies of the CLP red clay are, therefore, sensitive to summer monsoon precipitation variability.

In the late Miocene strata, alternations between brownish and light red palaeosols (Fig. 1b; Supplementary Fig. 1a) are associated with large-amplitude Al/Na, Rb/Sr, and lightness oscillations, which reflect orbital-scale summer monsoon intensity variations (Figs. 2–3). Spectral analyses of these alternations suggest the presence of strong

eccentricity (~100-kyr and 405-kyr) cycles (Fig. 3). Obliquity (41-kyr) cycles are relatively weaker in the Al/Na, Rb/Sr, and lightness records (Fig. 3). Notably, large-amplitude late Miocene-style ~100-kyr orbital variability becomes weaker in the early-to-middle Pliocene Al/Na, Rb/Sr, and lightness records, while eccentricity cycles remain dominant (Figs. 2–3). Furthermore, mean Al/Na and Rb/Sr increase markedly across the MPB, while lightness decreases substantially (Fig. 2b–d). These lithological and proxy changes provide strong evidence for increased summer monsoon precipitation and soil moisture availability across the MPB. Greater Pliocene soil moisture would have intensified chemical weathering and carbonate nodule leaching (which would have preferentially removed soluble carbonates), leading to a disappearance of late Miocene-style thick carbonate-nodule-rich light red palaeosols and instead to cumulative formation of dark red palaeosols, with only a few thin (0.2–0.4 m) carbonate nodule layers (Supplementary Fig. 1). Higher precipitation in association with increased summer monsoon moisture transport is a requirement for such enhanced pedogenesis and carbonate nodule leaching in particular, while increased temperature is not necessary. If a temperature increase is associated with a decrease in synchronous summer monsoon precipitation, the regional net moisture (precipitation minus evaporation) would decrease, which would weaken pedogenesis and carbonate nodule leaching on the CLP across the MPB, which contrasts markedly with observations from the Shilou and other red clay sections[17,19,26,30,32–38]. It appears that the CLP red clay Al/Na, Rb/Sr, and lightness records are more sensitive to changes in summer monsoon precipitation than temperature. We find that the summer monsoon precipitation increase on the CLP suggested by these red clay proxies coincided with a worldwide temperature increase across the MPB[6,7,39–41].

The Al/Na, Rb/Sr, and lightness records have similar orbital variability in the untuned magnetochronology as in our refined astronomical time scale, with prominent ~100-kyr and 405-kyr eccentricity cycles and relatively weaker obliquity cycles (Supplementary Fig. 5). In spectral analyses, we note that calculated eccentricity and obliquity bands are displaced slightly in the untuned magnetochronology, or that the obliquity expression is subdued in a few intervals where non-orbital signals appear to be more distinct. Orbital expression is enhanced, and non-orbital signals are subdued in the refined astronomical time scale relative to the untuned magnetochronology. These minor differences are consistent with the increased precision of the astronomical time scale, and do not influence our inference of monsoon variability, particularly summer monsoon intensification across the MPB, which is constrained by the palaeomagnetic polarity reversal boundary between the C3n.4n normal polarity chron and the C3n.4r reversed polarity chron.

Overall, the Shilou red clay lithological and proxy records suggest prominent orbital-scale monsoon variability throughout the late Miocene–middle Pliocene and a distinct transition to higher summer monsoon precipitation across the MPB. Some inevitable differences among the three Shilou red clay proxy records may be related to their diverse sensitivity to summer monsoon variability under different global climate conditions. For example, the Al/Na record has a larger amplitude increase in mean values across the MPB and lower amplitude orbital variability during the early-to-middle Pliocene than the Rb/Sr record, which may relate to enhanced Na leaching associated with intensified summer monsoon precipitation and chemical weathering.

## Discussion

In addition to the Al/Na, Rb/Sr, and lightness records from the Shilou section, the Jiaxian red clay Rb/Sr record (~120 km north of the Shilou section) also contains prominent 100-kyr and 405-

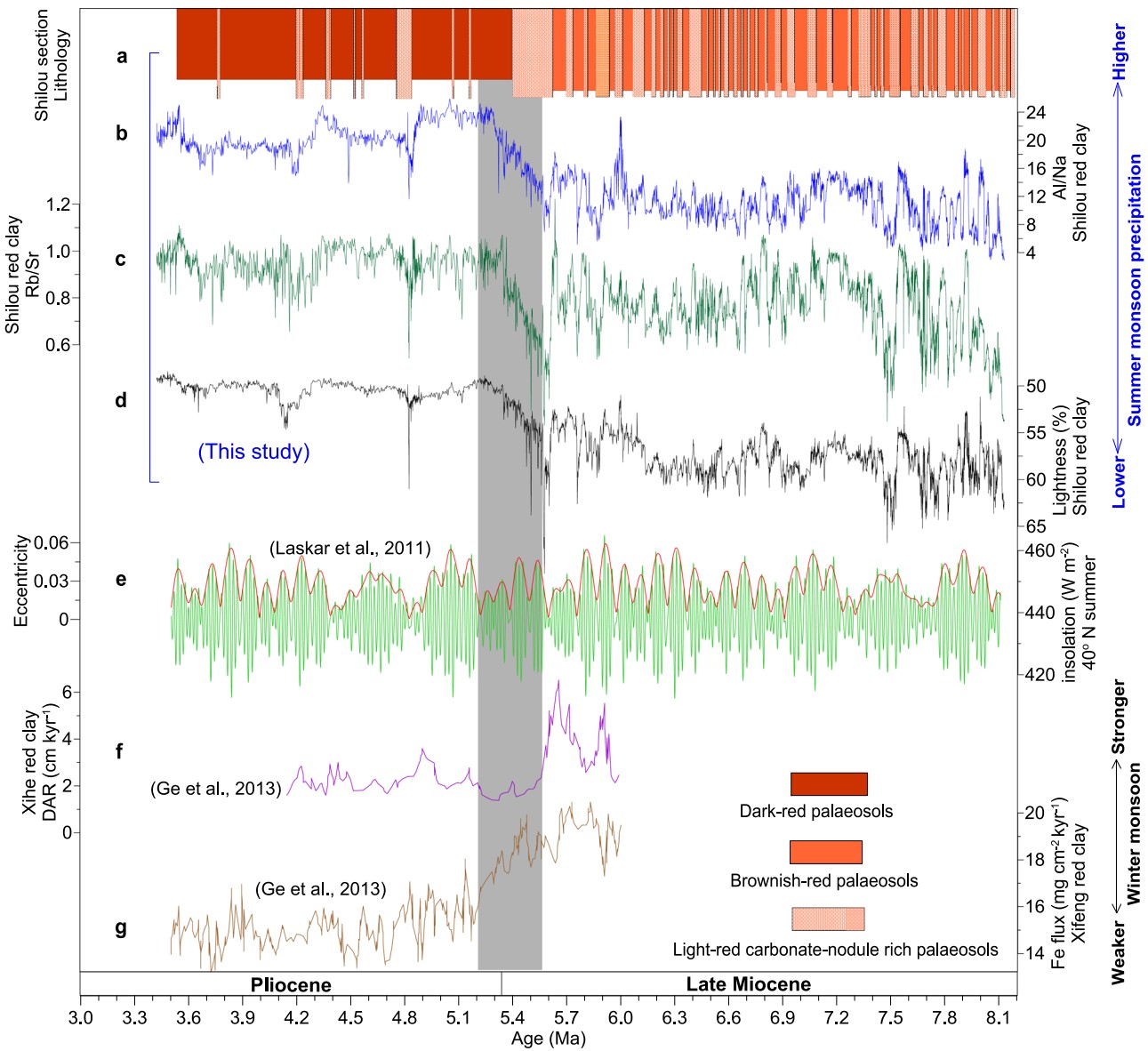

**Fig. 2 Lithology and palaeoclimate proxy data for red clay from the Chinese Loess Plateau. a** Lithology, **b** Al/Na, **c** Rb/Sr, and **d** lightness from the Shilou red clay sequence (this study). **e** 40° N summer insolation (green) and eccentricity (red) variations[31]. **f** Dust accumulation rates (DAR) for the Xihe red clay section[32]. **g** Fe flux for the Xifeng red clay section[32]. Summer and winter monsoon intensified and weakened, respectively, across the MPB (shaded).

kyr cycles[26] (Fig. 3d). The combined Shilou and Jiaxian records suggest that the Asian summer monsoon system varied dynamically over eccentricity timescales during the late Miocene–middle Pliocene. Eccentricity modulates the amplitude of summer insolation (Fig. 2e). Summer insolation is an important driver of Asian monsoon variability[42], so we infer that eccentricity modulation of summer insolation could have influenced the Asian monsoon during the late Miocene–middle Pliocene when the Northern Hemisphere was almost ice-free and global climate was warmer than today. It is likely that eccentricity modulation of summer insolation drove the observed distinct 405-kyr and 100-kyr Asian monsoon variability through non-linear interactions with the global carbon cycle. Similar transfer of insolation signals into global climate has been demonstrated in marine benthic δ[18]O and δ[13]C records[43,44]. In addition to pronounced eccentricity cycles, a moderate obliquity response is apparent in the Shilou records, which is expressed discontinuously in the lower-resolution Jiaxian red clay Rb/Sr record (Fig. 3). Such a response might be related to obliquity-induced

variations in Antarctic ice volume, sea level, and/or the thermal gradient between continental Asia and its surrounding oceans, which influence atmospheric heat and moisture transfer from the tropics to higher latitudes[45]. The moderate obliquity response is possibly due to the absence of large Northern Hemisphere ice sheets that could have amplified the climate response to obliquity forcing[11].

Speleothem δ[18]O records suggest that precession periodicity dominates summer monsoon variability in South China over the last 640 kyr[42,46]. The precession expression is significantly weaker than that of eccentricity and obliquity in the Shilou red clay Al/Na, Rb/Sr, and lightness records (Fig. 3), despite the fact that their ~1–2-kyr mean resolution is sufficient to robustly identify precession cycles. However, precession is expressed clearly in spectral analyses of these records after removal of the large-amplitude longer periodicities (>40-kyr, i.e. obliquity and eccentricity) that tend to overprint the precession signal (Supplementary Fig. 6). We only tuned the 100-kyr cycles; to avoid over tuning, we did not tune obliquity and precession cycles because they are weaker

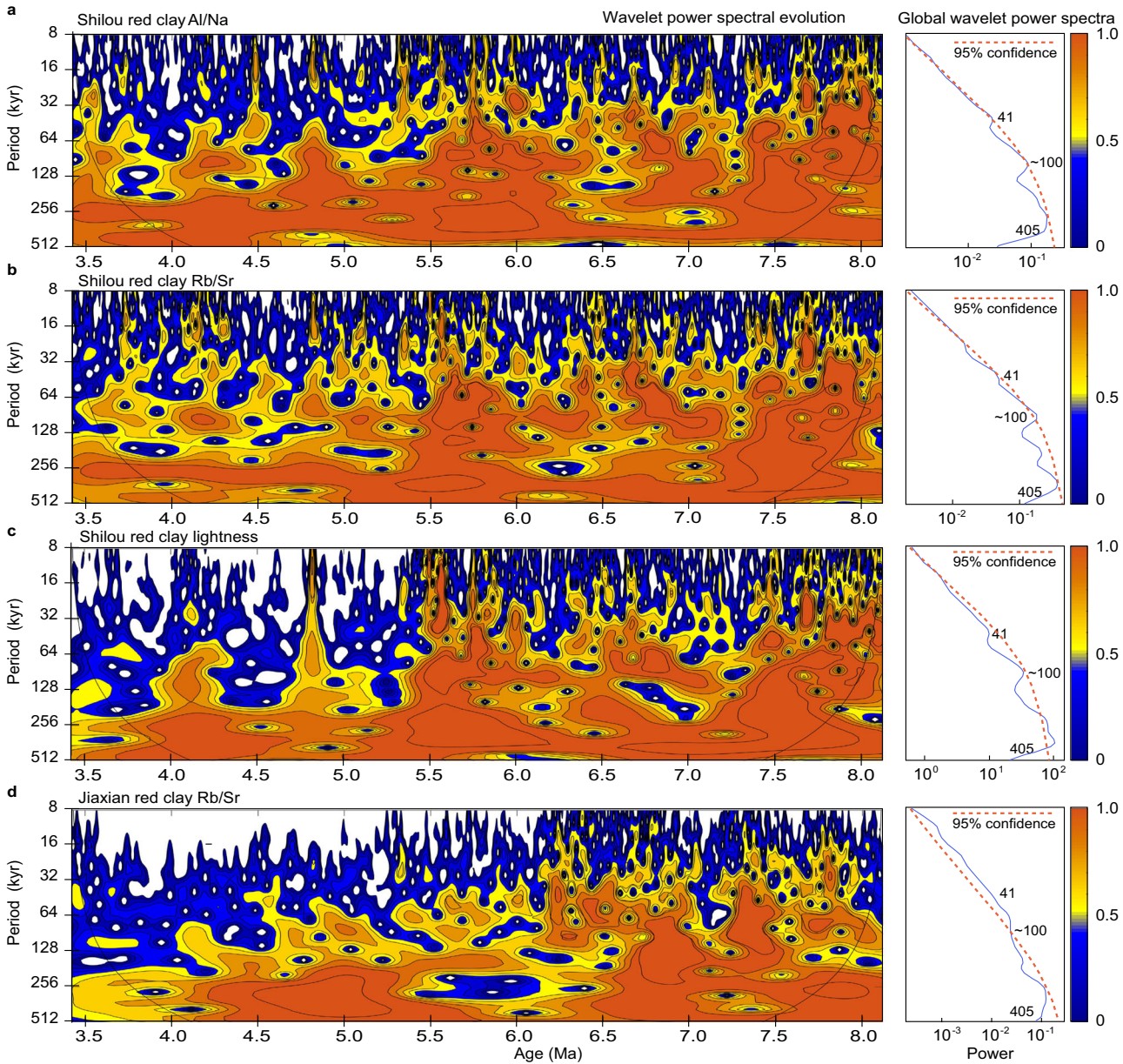

**Fig. 3 Orbital variability of Chinese Loess Plateau Asian summer monsoon records.** Wavelet power spectral evolution and global wavelet power spectra for **a** Al/Na, **b** Rb/Sr, and **c** lightness records from the Shilou red clay section, and **d** Rb/Sr record from the Jiaxian red clay section[26].

and less continuous than the 100-kyr variability (Supplementary Fig. 4). The weak precession expression possibly relates to strong smoothing under the warmer-than-present conditions with high atmospheric $CO_2$ concentrations, a mostly ice-free Northern Hemisphere, and low CLP red clay sedimentation rates. Under these conditions, thin originally less-weathered layers with low sedimentation rates deposited during precession/insolation minima may have been overprinted rapidly by intensified summer monsoon action and concomitant chemical weathering during the subsequent transition from insolation minima to maxima, with a downward-increased chemical weathering depth. This downward smoothing likely resulted in an attenuated expression of insolation minima (dry portions) and a preferential expression of maxima (wet portions) in the red clay summer monsoon proxy records, leading to a markedly subdued precession expression. This weak expression of (expected) precession cycles does not mean that the summer monsoon responded weakly to precession during the late Miocene–early Pliocene, but

that it is difficult to use these proxies to explore intricacies of monsoon responses to insolation over precession time scales due to recoding bias. Also, decreased 100-kyr amplitude variability across the MPB in the Shilou red clay Al/Na, Rb/Sr, and lightness records (Fig. 2), with concomitant weakening of short eccentricity (100-kyr) expression in spectral analysis (Fig. 3), is possibly linked to enhanced downward smoothing influence on the early-to-middle Pliocene red clay with increased temperature and summer monsoon precipitation, and decreased sedimentation rate, which might have weakened or blurred these orbital signals.

In addition to distinct orbital variability throughout the late Miocene–middle Pliocene, our red clay records suggest that the CLP climate changed prominently across the MPB, whereas orbital forcing did not change substantially (Fig. 2). We integrated our high-resolution records with existing palaeoclimate records from the wider Asian region to assess spatial climate changes across the MPB (Figs. 4–5; Supplementary Figs. 7–9). In contrast to our inference of a shift to a wetter CLP from the

Shilou red clay, many Central Asian arid regions beyond the reach of summer monsoon precipitation (Fig. 1a) experienced increased aridification across the MPB[47–50]. In the Tarim Basin, late Miocene lacustrine deposits with large-amplitude total organic carbon (TOC) and $CaCO_3$ content fluctuations that capture orbitally forced alternations between moist and dry conditions gave way to Pliocene fluvial-aeolian deposits with persistently low TOC and $CaCO_3$ contents[48,49] (Fig. 4a; Supplementary Fig. 7a). The associated decline in authigenic lacustrine carbonate formation led to decreased carbonate $\delta^{18}O$ (Fig. 4b) and increased $\delta^{13}C$ values[48,49] (Supplementary Fig. 7b). A shift to a less intense red colour (Supplementary Fig. 7c) and increased lightness (Fig. 4c) in the western Tarim Basin also indicates development of drier conditions across the MPB[49], in marked contrast to the shift to darker red colour (Fig. 1b) and decreased lightness (Fig. 2d) in the Shilou red clay, which indicate wetter climate as discussed above. A large free Fe (iron extractable by citrate-bicarbonate-dithionite) decrease in the northern Tarim Basin margin across the MPB (Supplementary Fig. 7d) points to less intense chemical weathering and enhanced aridification[49], which is supported by substantial increases in the dry-adapted herb taxa Artemisia and Chenopodiaceae in the neighbouring Junggar Basin[50].

We find corroborating evidence for increasing summer monsoon precipitation in the wider East Asian monsoon region across the MPB. For example, Jiaxian Rb/Sr values[26] increase and Xifeng and Xihe red clay ($CaO + Na_2O + MgO$)/$TiO_2$ and Na/K ratios[32] decrease across the MPB, which all suggest enhanced summer monsoon precipitation and stronger chemical weathering (Supplementary Fig. 8a–e). Many red clay $\chi$ records over the CLP are also characterised by a clear increase across the MPB[19,30,33–37] (Supplementary Fig. 8f–k), which is consistent with increasing precipitation. A sharp shift to redder colouration across the MPB is observed not only in the Shilou section (Fig. 1b), but also in numerous other sections across the CLP, such as at Xifeng[38], Jiaxian[35], Baode[34], Pianguan[33], and Linxian (Supplementary Fig. 9). The colour shift again indicates summer monsoon intensification across the MPB. In central Japan, increasing moist-adapted Cathaya (pine family) and warm-temperate pollen types across the MPB imply a shift to warmer and wetter conditions[51]. In the South China Sea and the Andaman Sea, negative shifts of seawater and planktic $\delta^{18}O$ (Fig. 4d) at Ocean Drilling Program (ODP) Site 1146 and International Ocean Discovery Program (IODP) Site U1448 across the MPB indicate increasing summer monsoon precipitation[7,40]. Parallel planktic Mg/Ca records at these two locations indicate that the hydroclimate changes were associated with a > 3 °C sea surface temperature (SST) increase[7,40]. Such a large temperature increase is expected to have driven substantial precipitation increases in East Asia, which is consistent with our observed prominent Al/Na and Rb/Sr increases and decreased lightness in the early-to-middle Pliocene Shilou red clay (Figs. 2 and 4e). In addition, the South China Sea $\chi$ and anhysteretic remanent magnetisation (ARM) records also suggest that higher summer monsoon precipitation increased terrigenous input across the MPB[52]. A few other proxy records from the southern CLP margin (Weihe Basin) and the South China Sea do not suggest summer monsoon intensification across the MPB[14,53–55], which seems to differ from our Shilou and the numerous discussed records from the wider East Asian monsoon regions. These differences, however, might be a result of the different proxies and locations. Red clay sections close to the northern or western summer monsoon boundary could be more sensitive to capturing monsoon changes across the MPB than records from the southern CLP and South China Sea, which are located well within the summer monsoon region. Some chemical weathering records from the South China Sea may not only express summer monsoon variations, but also changes in oceanic conditions and post-depositional diagenesis[14], which may have overprinted the summer monsoon intensification across the MPB. Future development of high-resolution records of the same proxies from different red clay sections, and simultaneous generation of different sensitive summer monsoon proxy records without non-monsoonal interferences from the same section, are pertinent to investigate such spatial variations in greater depth.

The observed summer monsoon intensification across the MPB is likely associated with a synchronous winter monsoon weakening because the average sedimentation rate in the Shilou red clay drops from ~2 cm/kyr in the late Miocene interval to ~1 cm/kyr in the early-to-middle Pliocene interval. Likewise, dust accumulation rates and Fe and Al fluxes of aeolian red clay decreased across the MPB[17,32,35] (Figs. 2f–g and 5a). These changes suggest a decline in Pliocene winter monsoon dust transport from arid northwestern China to the CLP compared to the late Miocene. Si/Al and Zr/Rb correlate positively with red clay grain size variations that relate to winter monsoon dust transport[20,26]. A transition to finer red clay particles across the MPB indicated by decreased Si/Al and Zr/Rb is supported by direct grain size measurements, which suggest smaller median grain sizes, increased fine (5–16 µm) particle concentrations, and decreased coarse (>20 µm) particle concentrations during the Pliocene relative to the late Miocene[20,26,56] (Fig. 5b–f). Grain size is routinely used as a winter monsoon proxy for both Quaternary loess and underlying late Miocene–Pliocene red clay[17,20,35,56], so its shift across the MPB stems dominantly from winter monsoon weakening, which drives decreased coarse dust transport to the CLP. Summer monsoon strengthening, which facilitates pedogenic fine particle formation, may also decrease red clay grain size. However, its influence is deemed to be smaller than the impact of dust accumulation variations because pedogenic clay represents a substantially lower portion of the strata compared to wind-blown red clay accumulation[15]. Another interpretation that relates the decreased dust grain size and accumulation rate of the Pliocene red clay to wetting in Central Asia (the most prominent dust source region) is unlikely because this region experienced drier climates according to numerous aforementioned robust records (Fig. 5a–c; Supplementary Fig. 7). In addition, the late Miocene and early Pliocene Dongwan red clay sequence (western CLP) is dominated by cold/dry-adapted and warm/humid-adapted terrestrial mollusks, respectively, which again indicate winter monsoon weakening and summer monsoon strengthening across the MPB[57] (Fig. 5g–h).

Synthesising these observations, we infer that Asian hydroclimate changes across the MPB coincided with global changes but with distinctly asymmetric spatial expression (Figs. 4–5; Supplementary Figs. 7–9). Reconstructed $CO_2$ records indicate that atmospheric $CO_2$ levels increased by ~100–250 ppm from the late Miocene (~250 ppm) to the early Pliocene (~320 to ~470 ppm)[8–10] (Fig. 4f). Increased atmospheric $CO_2$ concentration is supported by a prominently decreased $\delta^{13}C$ gradient between South China Sea planktic and benthic foraminifera[7] (Fig. 4g). At the same time, widespread ocean warming occurred: SST increased in the South Pacific Ocean, South Atlantic Ocean, Norwegian Sea, equatorial Pacific Ocean, subtropical eastern Indian Ocean, and South China Sea;[6,7,39] benthic $\delta^{18}O$ decreased in the Pacific and Atlantic Oceans;[7,41] and sea level rose by ~20 m in response to a major Antarctic ice-sheet decay[5] (Fig. 4h–n). It appears that global climate shifted prominently across the MPB, while orbital forcing did not change markedly. Thus, we suggest that our observed synchronous rapid continental-scale Asian hydrological gradient intensification across the MPB was not driven by orbital forcing, but was related primarily to coeval global warming, as further supported by our model simulations presented below.

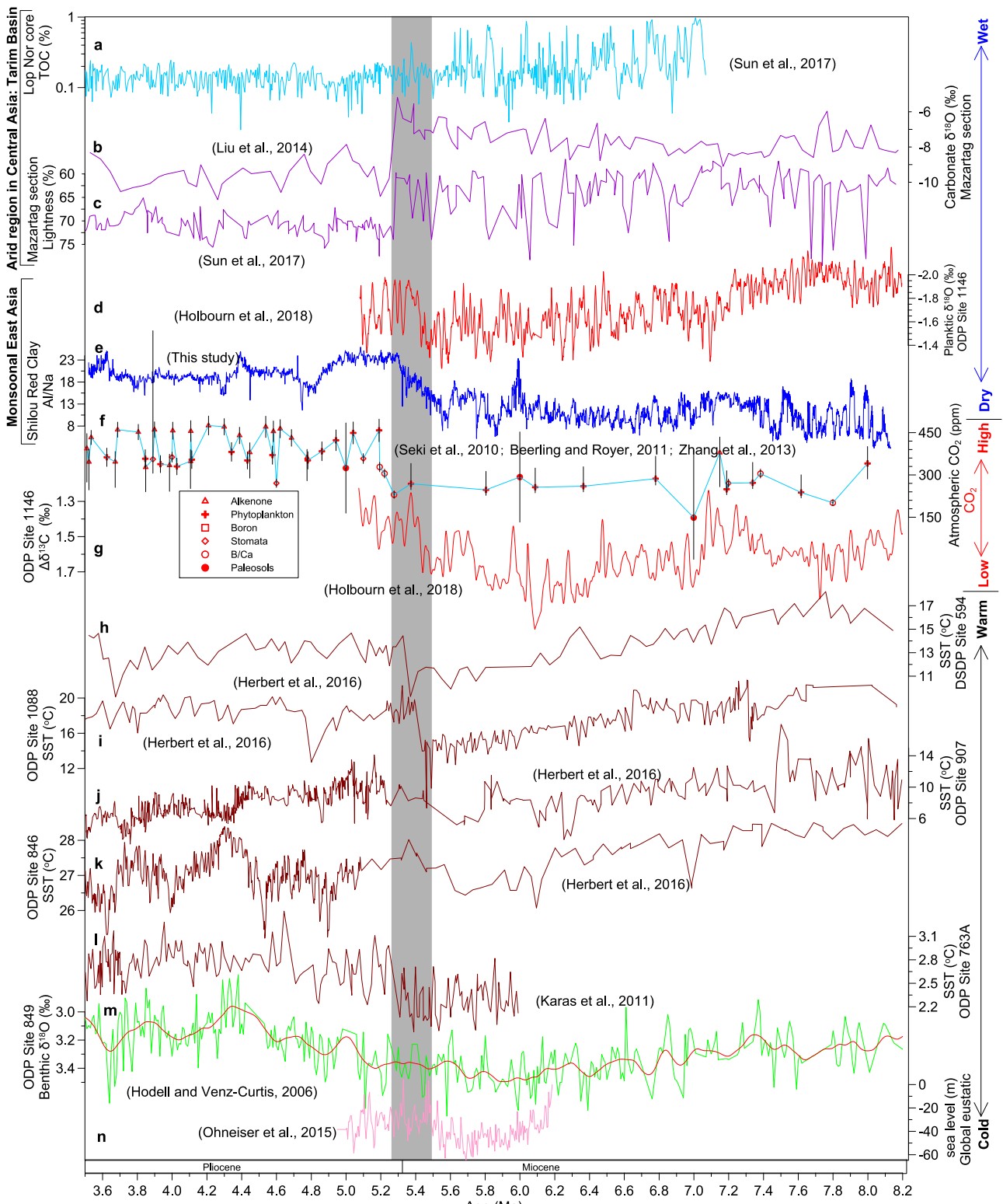

**Fig. 4 Compiled records of Asian and global climate change across the MPB. a–c** Total organic carbon (TOC), carbonate δ[18]O, and lightness records from the Tarim Basin[48,49]. **d** ODP Site 1146 planktic δ[18]O record[7]. **e** Al/Na from the Shilou red clay section (this study). **f** Compiled proxy record of atmospheric $CO_2$ concentrations from different methods, with error bars representing one-sigma uncertainties[8–10]. **g** Gradient between planktic and benthic foraminiferal δ[13]C at ODP Site 1146, South China Sea[7]. **h–l** Sea surface temperature (SST) reconstructions[6,39] from DSDP Site 594 (South Pacific Ocean) and ODP Sites 1088 (South Atlantic Ocean), 907 (Norwegian Sea), 846 (tropical eastern Pacific Ocean), and 763 A (subtropical southeastern Indian Ocean). **m** ODP Site 849 benthic foraminiferal δ[18]O record, tropical eastern Pacific Ocean[41]. The red smoothed curve is fitted using the locally weighted least squares error (LOWESS) method with Matlab. **n** Modelled global eustatic sea level changes[5]. The compiled data indicate increased moisture in Asian monsoon areas and aridification in arid Central Asia, as $CO_2$ and temperature increased across the MPB (shaded).

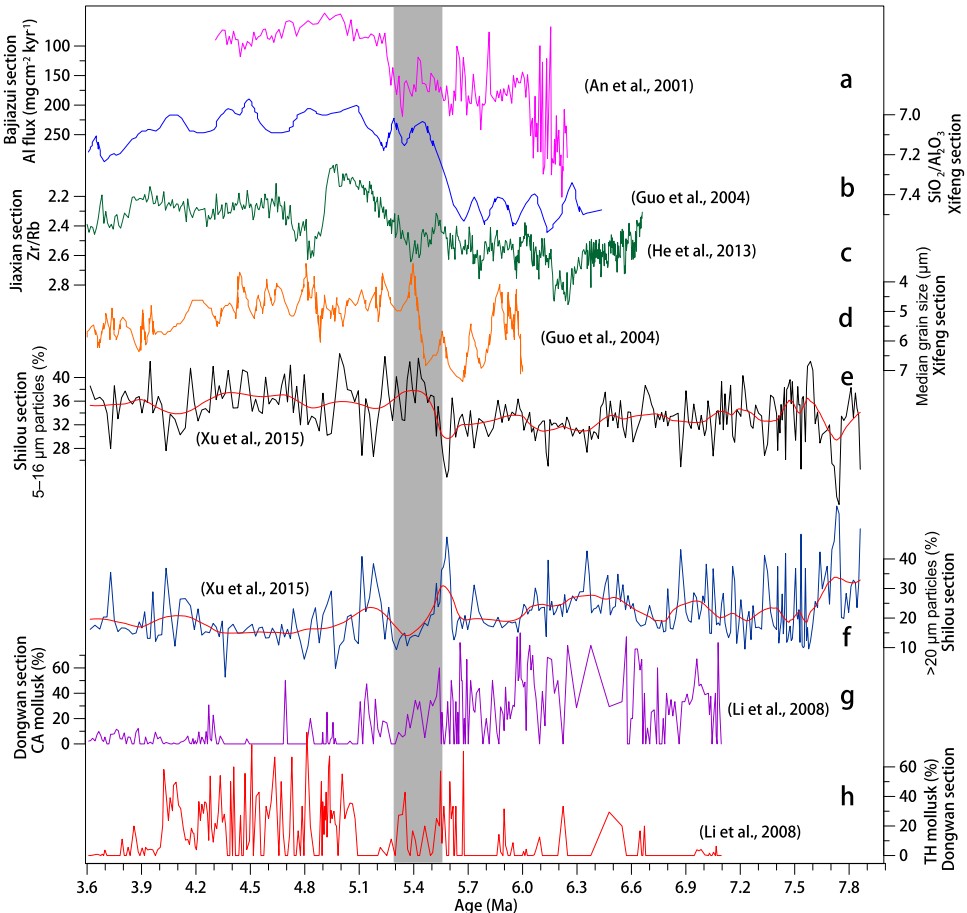

**Fig. 5 Chinese Loess Plateau red clay records of climate variability. a** Al flux at the Bajiazui section[17]. **b** $SiO_2/Al_2O_3$ from the Xifeng section[20]. **c** Zr/Rb from the Jiaxian red clay section[26]. **d** Median grain size from the Xifeng section[20]. **e, f** Fine (5–16 μm) and coarse (>20 μm) particle concentration from the Shilou section[56]. Red smooth curves are fitted using the MatLab LOWESS function. **g, h** Cold-aridiphilous (CA) and thermo-humidiphilous (TH) terrestrial mollusk changes in the Dongwan red clay[57].

To investigate Asian hydroclimate responses to $CO_2$-induced global warming across the MPB, we used the Community Earth System Model (CESM 1.0.4)[58] to perform two simulations for late Miocene and Pliocene conditions, respectively (see "Methods" for details). The Pliocene simulation was conducted with Pliocene boundary conditions from the PRISM4 (Pliocene Research, Interpretation and Synoptic Mapping version 4) dataset[59], including Pliocene orography, vegetation, ice sheets, lakes, and 400 ppm $CO_2$ concentration, along with modern orbital parameters, modern solar constant, and preindustrial $CH_4$, $N_2O$, and aerosol conditions. The late Miocene simulation was also conducted with the same PRISM4 boundary conditions but with a lower $CO_2$ level of 280 ppm. Similar to the present day (Fig. 1a; Supplementary Fig. 10), the late Miocene and Pliocene simulations both produce pronounced monsoonal winds and summer rains over South and East Asia, with arid conditions in Central Asia (Fig. 6a–b). Summer temperature is higher in the higher-$CO_2$ Pliocene simulation and increases more over higher latitudes than over East Asian monsoon regions (Fig. 6c). From the late Miocene experiment to the Pliocene experiment, summer precipitation increases over most of Asia, while it decreases over the northern Tibetan Plateau (Fig. 6d). Net summer (Fig. 6e) and annual (Supplementary Fig. 11a) surface moisture (precipitation minus evaporation) changes have a similar spatial pattern with a wetter East Asian monsoon region and a drier Central Asia. These moisture changes are consistent with strengthening of summer monsoon circulation and atmospheric ascent in East Asia

(Fig. 6f–g) and increased lower-tropospheric water vapour loading in the Western Pacific and Indian Oceans due to warmer conditions in the Pliocene experiment (Fig. 6c). Also, associated enhanced cyclonic deviation circulation around the Tibetan Plateau "pumps" surrounding flows (including the summer stream field at 850 hPa) to converge into the Tibetan Plateau[60]. These changes together result in increased warm/moist southerly flow from the Western Pacific and Indian Oceans and subsequently more rainfall in East Asian monsoon regions (Fig. 6d), in a pattern that is consistent with modern projections and observation-based analysis[61–63]. In addition, our simulations indicate that the summer westerly jet (Fig. 6h) and winter monsoon winds (Supplementary Fig. 11b) become weaker in response to a warmer high-latitude Northern Hemisphere, which facilitates inland penetration and prolongs summer monsoon duration, resulting in a wetter East Asia[64]. Based on those simulation results, we suggest that $CO_2$-induced MPB warming could have driven the coeval shift to wetter conditions in monsoonal East Asia.

In both late Miocene and Pliocene simulations, Central Asia is largely beyond the reach of summer monsoon precipitation (Fig. 6a, b). Thus, warming-induced increased summer moisture supply and transport cannot penetrate effectively to this remote inland region, which is bounded by high-altitude mountain ranges. In this region, low precipitation (generally <100 mm/yr) is associated with strong evaporation. Our simulations suggest that summer precipitation does not decrease to the same extent

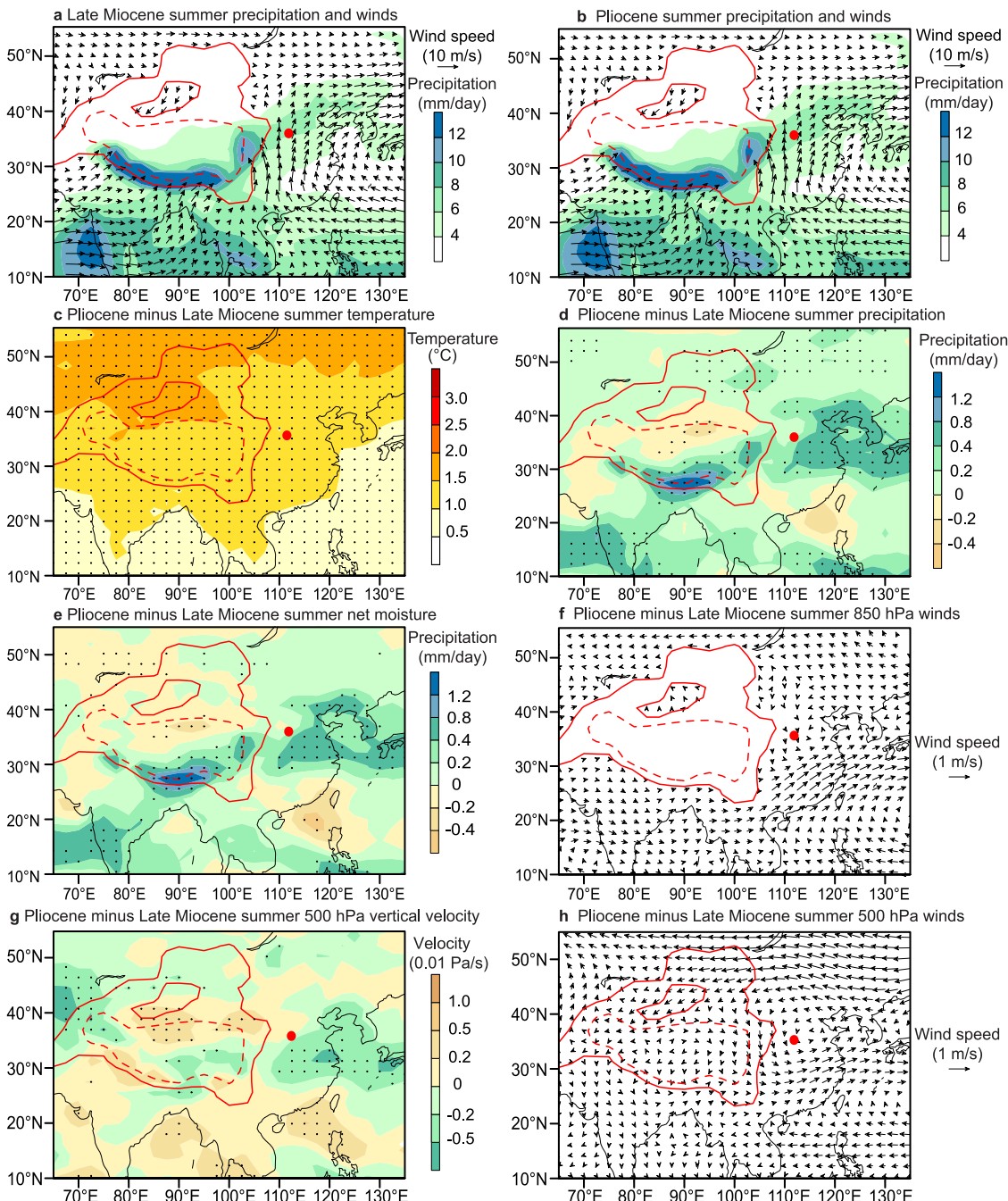

**Fig. 6 Simulated Asian climate and atmospheric circulation responses to CO₂ increase across the MPB.** Simulated summer precipitation (shading) and 850 hPa winds (black vectors) for **a** late Miocene (280 ppm CO₂) and **b** Pliocene (400 ppm CO₂) experiments. Simulated changes (Pliocene minus late Miocene) of summer **c** surface air temperature, **d** precipitation, **e** net moisture (precipitation minus evaporation), **f** 850 hPa winds (summer monsoon), **g** 500 hPa vertical velocity, and **h** 500 hPa winds (westerly jet), due to CO₂ increase from the late Miocene level of 280 ppm to the Pliocene level of 400 ppm. Solid and dashed red inland contours in (a–h) denote 1500 and 3000 m topographic contours, respectively. Negative (green) and positive (yellow) shades in **g** represent increasing atmospheric ascending and descending flows, respectively. Note that stronger atmospheric ascending and descending flows generally result in more and less precipitation, respectively. The red dot indicates the location of the Shilou section. Black dots in **c–e** and **g** denote regions statistically significantly above the 95% confidence level (Student's *t*-test). Summer is represented by May to September.

everywhere across Central Asia from the late Miocene experiment to the Pliocene experiment (Fig. 6d–e). Particularly, Central Asia has higher summer precipitation across the ~40–55°N latitudinal band (Fig. 6d), but still becomes drier (Fig. 6e) due to the large temperature increase and strong evaporation. In addition, warming-induced intensification of spiral air pumping tends to enhance anticyclonic deviation circulation gyres at high latitudes and facilitate high-latitude cold/dry air advection across Central

Asia[60]. Thus, the MPB warming could have enhanced soil water evaporation to exacerbate the net moisture deficit (precipitation minus evaporation) over Central Asia, leading to intensified aridification in much of Central Asia as observed from aforementioned palaeoclimate reconstructions.

Synchronous summer monsoon strengthening, winter monsoon weakening, and decreased dust transport to the CLP[7,17,20,57] (Figs. 2 and 5) seem to be incompatible with a response to

regional uplift of the northern Tibetan Plateau across the MPB, given that such uplift would instead intensify the winter monsoon[17] and increase cold/dry equatorward flow from high- to low-latitude Asia, resulting in colder and drier winters[60] and increased CLP aeolian mass accumulation rates. Moreover, tectonic processes associated with uplift are anticipated to have occurred diachronously across various regions, in a protracted process over millions of years. In contrast, our synthesis demonstrates that East and Central Asian climate conditions shifted across the MPB synchronously and rapidly within a few short-eccentricity cycles. Finally, evidence increasingly suggests that the Tibetan Plateau was already at high altitudes in the late Miocene or in the early to middle Miocene[65–69]. Only small-scale regions of the Tibetan Plateau and adjacent mountain ranges were uplifted during the late Miocene or Pliocene, as inferred from northern and eastern Tibetan Plateau sedimentary records[70–73], minor tectonic deformation in the Pamir and Tian Shan mountains[74–78], and low-temperature thermochronology of the Shanxi Rift (North China), northern Tibetan Plateau, and Mongolian mountains[79–86]. Therefore, we consider regional uplift to have played a secondary role in Asian climate change across the MPB.

To summarise, multiple high-resolution proxy records from red clay on the CLP suggest that the Asian summer monsoon responded dynamically to 405-kyr and 100-kyr eccentricity and moderately to 41-kyr obliquity between 8.1 and 3.4 Ma when atmospheric $CO_2$ concentrations were high and the Northern Hemisphere was largely ice-free. Furthermore, parallel lithological and proxy records suggest distinct summer monsoon intensification across the MPB. Combining existing records, we find a hydrological gradient intensification over Asia across the MPB: East Asia became wetter, while many arid Central Asian regions became drier, with accompanying biogeographic and evolutionary implications. Our model simulations demonstrate that this large-scale Asian hydroclimate change was linked to global warming across the MPB that was driven by higher Pliocene $CO_2$ concentrations. Following this transition, Asian hydroclimate did not recover back to its late Miocene configuration, even though temperatures decreased from the early to middle Pliocene onward. Based on our observed global and regional changes across the MPB from palaeoclimate reconstructions and simulations, we posit that continued hydrological gradient intensification may be expected over Asia under anthropogenic global warming. Our findings provide palaeoclimatic support for projections of future regional hydrological trends under long-term sustained anthropogenic emissions: Central Asia will likely become even drier than it is currently, with more persistent droughts and enhanced desertification[61,87]. Meanwhile, most East Asian monsoon regions will likely become even wetter, with increased flooding risks[61,62]. Such Asian climate changes will have substantial impacts on regional irrigation systems, ecosystems, and society.

## Methods

After removal of the weathered outcrop surface, 3,527 fresh samples were collected from the late Miocene to middle Pliocene red clay at 2-cm intervals (corresponding to a ~1–2 kyr time spacing) from the Shilou red clay section, eastern Chinese Loess Plateau (CLP). All collected samples were used for the following palaeoclimate proxy measurements. About 5 g of sediment was dried at 40 °C for 24 h and ground to <38 μm with an agate mortar and pestle. Powders were compacted into oblate discs (32-mm diameter) enclosed by a polyethylene ring with a tablet machine. Disc samples were used to determine bulk element concentrations using an Axios advanced wavelength dispersive X-ray fluorescence (XRF) instrument (WD-XRF; PANalytical, Almelo, The Netherlands). The relative standard deviation, based on repeated analyses of National Standards GSS-8 and GSD-12, was <2% for all major elements (Al and Na) and ~10% for all trace elements (Rb and Sr). After XRF analysis, samples were used for colour reflectance measurements with a Minolta CM-508i Spectrophotometer. Results are expressed as the spherical L*a*b* colour

space and reflectance intensity between 400 and 700 nm. L* describes the lightness between black (0) and white (100), while a* and b* denote red–green and yellow–blue chromaticity, respectively. All proxy measurements were carried out at the Institute of Earth Environment, Chinese Academy of Sciences, Xi'an, China.

We used an automatic orbital tuning approach[88] and the Acycle software[89] to generate an astronomical time scale and to evaluate orbital signatures, respectively. Based on the most recent magnetostratigraphic data for the Shilou red clay section[4,19,30], we first established an untuned magnetochronology through linear interpolation using geomagnetic polarity reversals for age control, assuming constant long-term sedimentation rates between reversals (Supplementary Fig. 3). We then conducted spectral analyses of the Rb/Sr record in the untuned magnetochronology, which suggest a continuous 100-kyr eccentricity band throughout the late Miocene–Middle Pliocene, with weaker and less continuous obliquity and precession cycles (Supplementary Fig. 4a). Accordingly, we tuned large-amplitude 100-kyr Rb/Sr variations to the astronomical solution[31] to achieve cycle-to-cycle correlation within magnetochronological constraints. We filtered the 100-kyr component from the Rb/Sr record using a 90–125 kyr Gaussian bandpass filter with the Acycle software[89], with 0.009 cycle/kyr centre and 0.001 cycle/kyr bandwidth. Ages for palaeomagnetic reversals were not kept fixed to optimise tuning results given uncertainties in palaeomagnetic boundaries (10–60 cm; Supplementary Table 1), post-depositional natural remanent magnetisation (NRM) lock-in depth in aeolian sediments[90,91], and GPTS ages[92]. Generally, high Rb/Sr and filtered 100-kyr Rb/Sr peaks were associated with wet climates and are correlated to eccentricity maxima when insolation was high. After checking >30 different correlations between the filtered 100-kyr Rb/Sr component and 100-kyr eccentricity[31], we found that three options (youngest, intermediate, and oldest tuned age models) resulted in synchronous cycle-to-cycle correlations of the filtered 100-kyr Rb/Sr component with orbital 100-kyr eccentricity in both coherency and amplitude modulation patterns, with reasonable sedimentation rate changes and broadly consistent palaeomagnetic reversal ages, within uncertainty of their GPTS ages (Supplementary Fig. 3; Supplementary Table 1). The three options were established using 50 age correlation points where Rb/Sr minima facilitated consistent correlation point selection throughout the entire interval (Supplementary Fig. 3; Supplementary Table 2). Orbitally tuned age models should vary between the youngest and oldest tuned age models, so the 50 selected tie points were moved largest backward (younger limit) and forward (older limit) to produce the youngest and oldest tuned age models. Along with the same tuning strategy, which integrated eccentricity correlations, sedimentation rates, and magnetochronology, the intermediate age model levelled out between the youngest and oldest tuned age models (Supplementary Fig. 3; Supplementary Table 2). In the intermediate age model, the filtered 100-kyr Rb/Sr component correlates cycle-to-cycle with orbital eccentricity in both coherency and amplitude modulation patterns, and sedimentation rates vary gradually (without sharp changes that are unlikely for an aeolian lithology), but palaeomagnetic reversal ages are more consistent with their GPTS ages than in the youngest and oldest tuning age models (Supplementary Table 1). Thus, the intermediate age model is the most consistent option, and was selected as the final astronomical age model for the Shilou red clay sequence. Ages of the 50 tie points in the intermediate tuning option minus their ages in the youngest and oldest options were used to estimate potential negative and positive age uncertainties, respectively (Supplementary Fig. 3; Supplementary Table 2). Despite uncertainties in tuning (Supplementary Table 2), palaeomagnetic reversal depths (Supplementary Table 1), post-depositional NRM lock-in, and GPTS ages, all three orbitally tuned age models and the untuned magnetochronology produce similar major spectral evolutionary Rb/Sr features, with predominant 405-kyr and ~100-kyr periodicities throughout the ~8.1–3.4 Ma interval (Supplementary Fig. 4). This suggests that the prominent eccentricity expression in the Shilou red clay sequence is a robust feature rather than an orbital tuning artefact. Our minimal 100-kyr eccentricity tuning strategy without further tuning with 41-kyr and ~20-kyr cycles (because of their weak and less continuous expression) decreases the risk of over-tuning.

The Community Earth System Model (CESM 1.0.4)[58] was used to elucidate underlying dynamics of the observed contrasting hydroclimate change in the East Asian monsoon region and arid Central Asia across the MPB (Fig. 6; Supplementary Figs. 10–11), and particularly to test its sensitivity to coeval $CO_2$-induced global warming under late Neogene boundary conditions. CESM is a widely used global model with coupled dynamic atmosphere, ocean, land, and sea-ice components. The atmosphere has a spatial resolution of ~1.9° (latitude) × 2.5° (longitude) in the horizontal direction, and 26 layers in the vertical direction. Land has the same horizontal resolution as the atmosphere. Ocean and sea-ice components have a ~1° horizontal resolution. The ocean component contains 60 vertical layers. First, following guidelines of the Pliocene Model Intercomparison Project phase 2 (PlioMIP2)[93], we conducted a simulation with preindustrial boundary conditions, including modern orography, vegetation, ice sheets, lakes, and 280 ppm atmospheric $CO_2$ concentration. The simulated preindustrial Asian summer precipitation and circulation pattern is similar to the observed present-day pattern, with higher precipitation and stronger summer monsoon in South and East Asia than in northwestern Asia (Supplementary Fig. 10). This indicates that CESM 1.0.4 can reproduce present-day climate features[58] and is useful for simulating late Neogene Asian palaeoclimate[94,95].

Based on recent modelling of the global monsoon responses to Pliocene boundary conditions[95], two other numerical experiments were performed in detail

to evaluate climatic responses of monsoonal Southeastern Asia and arid Central Asia to $CO_2$-induced global warming across the MPB. One scenario represents Pliocene conditions and was conducted with reconstructed Pliocene orography, ice sheets, vegetation, lakes, and a 400 ppm atmospheric $CO_2$ level from the PRISM4 (Pliocene Research, Interpretation and Synoptic Mapping version 4) dataset[59], with modern orbital parameters (year 1950), modern solar constant (1365 W/m$^2$), and preindustrial $CH_4$, $N_2O$, and aerosol conditions. The other scenario represents late Miocene conditions and was conducted with the same boundary conditions, except for a lower $CO_2$ level of 280 ppm. The Pliocene simulation was initiated from the default preindustrial simulation (including deep ocean temperature) and was run for 2,050 model years. The late Miocene simulation was initiated from model year 1501 of the Pliocene experiment and was run for a further 550 model years. The atmosphere and upper ocean in both experiments reached quasi-equilibrium. In the Pliocene Model Intercomparison Project, most Pliocene simulations were run for 500 or >500 model years and the last 30, 50, or 100 model years were generally used to calculate climatological means[93,96]. We, therefore, analysed the climatological means of the last 100 model years from each experiment. Application of Pliocene surface boundary conditions is an improvement from previous simulations of Asian climate responses to $CO_2$-induced global warming, which typically used preindustrial/modern boundary conditions[61,62,87,97–99]. Compared to the preindustrial, the late Neogene was warmer and wetter with smaller ice and snow cover (largely ice-free Northern Hemisphere), larger vegetation cover, and lower meridional and zonal temperature gradients[59,100,101]. Asian climate might respond more significantly to simulated $CO_2$ increases with present-day boundary conditions than to late Neogene boundary conditions because of enhanced feedbacks under pre-industrial conditions (e.g., larger snow/ice and smaller vegetation cover). Consistent with previous preindustrial/modern boundary condition simulations[61,62,87,97–99], however, our simulations with late Neogene boundary conditions also suggest that $CO_2$-induced global warming can induce wetter conditions in the East Asian summer monsoon region and drier conditions in Westerly-dominated arid Central Asia, which support the 'wet-gets-wetter and dry-gets-drier' projections for Earth's climate system under sustained anthropogenic forcing[61]. Asian climate responses to $CO_2$-induced global warming appear to follow a broadly similar pattern under different late Neogene and present-day boundary conditions. Thus, uncertain late Neogene boundary conditions (e.g., topography, greenhouse gas concentrations, oceanic thermal and circulation conditions, and atmospheric circulation) could result in subtly different simulations, but the general Asian climate response pattern to $CO_2$ increase under adjusted late Neogene boundary conditions seems robust: it does not change substantially.

## Data availability

All of our measured proxy data presented here are attached in the Supplementary Dataset 1 and are also available in the East Asian Paleoenvironmental Science Database (http://paleodata.ieecas.cn/index.aspx, Data https://doi.org/10.12262/IEECAS.EAPSD2021004). The model simulation data used to support our dynamic interpretations are available at: https://zenodo.org/record/4964420.

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

## Acknowledgements

This study was supported financially by the Chinese Academy of Sciences (CAS) Strategic Priority Research Program (XDB 40000000), the Second Tibetan Plateau Scientific Expedition and Research (STEP) program (2019QZKK0707), the Chinese Academy of Sciences Key Research Program of Frontier Sciences (QYZDB-SSW-DQC021), the National Natural Science Foundation of China, the Ministry of Science and Technology of China, Australian Research Council (ARC) Australian Laureate Fellowship grant FL120100050 to E.J.R., ARC grant DP120103952 to A.P.R., ERC consolidator grant MAGIC 649081 to G.D.-N., and Heising-Simons Foundation Grant #2016-05 to C.J.P.

## Author contributions

H.A. conceived the idea of this study. H.A., F.W., X.K.Q., and P.Z. performed multi-proxy measurements. R.Z. performed the modelling simulations. Z.A., E.J.R., A.P.R., A.E.H., G.D.-N., W.K., M.J.D., Q.S.L., Z.H.L., A.L., W.J.Z., Z.D.J., and Y.X. contributed to proxy analysis, interpretation, and discussion. J.-B.L., C.J.P., J.C., X.D.L., G.X.W., and X.Z.L. contributed to modelling design, analysis and discussion. Q.S., C.M., X.X.L., and X.Z.P. helped with orbital tuning and spectral analysis. H.A. led the manuscript writing with intellectual contributions from all authors.

## Competing interests

The authors declare no competing interests.
