## [Peer Review File · Nature Communications]

Global warming-induced Asian hydrological climate transition across the Miocene–Pliocene boundaryReviewers' Comments:

Reviewer #1:

Remarks to the Author:

The authors present the proxy records from Shilou aeolian red clay on the Chinese Loess Plateau. The data comprise the past period from 8.1 ma to 3.5 ma, across the late Miocene to mid-Pliocene, indicate an apparent transition occurred around Miocene-Pliocene Boundary. According to the the red clay's response to climate, the authors suggest that the proxy records inferred changes in regional climate in response to increased CO₂ from Miocene to Pliocene, implies a similar climate response pattern will occur in the future anthropogenic GHG forcing . The new dataset provide robust and solid evidence on local hydro climate transition from late-Miocene to mid-Pliocene.

Data site Shilou locates at the northern boundary of East Asian monsoon, is sensitive to the changes in east Asia monsoon. The authors suggest that changes in this data can well represent the changes in east Asian monsoon. By including several other published proxy data, the authors also show a regional climate change pattern and aim to provide the paleoclimate evidence to support the "wet gets wetter and dry gets drier" global climate change in the future. The work is a valuable contribution to infer the consequence of ongoing global warming. I have some comments for authors to consider and revise the manuscript.

P3. L45, also in P4 L60, About the Miocene-Pliocene boundary, when checking in the literature, the definition for the Miocene-Pliocene boundary has been mentioned frequently, but no literature define the MPB at ~5.3 ma. If it is a new definition by this study, should clarify. Otherwise it would be misleading.

When presenting the data, the authors have had spectrum analysis and discussed the orbital variability in length. It is not very clear the role of orbital forcing in the MPB transition. My understanding is that the authors try to explain that the MPB transition is not due to the orbital forcing. If this is the case, considering most of the Nature Communication readers may not understand these forcing processes, this section can be largely shorten to avoid the confusion.

P8 L73, What is "C3n.4n-C3r polarity reversal boundary" ? It is hard to follow.

P23, L 275-285, The authors used oceanic proxy data in Indian Ocean to show the intensified summer monsoon, SSTs in Indian Ocean often affect the South Asian monsoon, when SST increase in these regions, it will decrease the land-sea contrast and weaken southerly wind, and weaken the South Asian monsoon (Roxy, et al., 2015), may provide less moisture to East Asia as well. I suggest the authors to discuss the changes in the land-sea contrast carefully.

In the climate modelling part, the only difference in the simulations for Miocene and Pliocene is CO₂ concentration (280 vs 400), this is similar to the present condition except the different boundary condition setting by PRISM4. Some discussion on the effect of the PRISM4 boundary condition is needed to clarify the climate response to CO₂ during MPB and present day.

Burls and Fedorov (2017) used the same model CESM and did the simulation for Pliocene , they have drawn the opposite conclusion. Authors may compare with their work and discuss the possible reason why the same model simulate the opposite results.

The climate responses showed in Fig6 are based on the 30 years simulation data, to represent the climate change on million years time scale. Considering the well know multidecadal and centennial climate variability is present in both proxy data and model simulations, using 30 years mean is not enough. According to the model experiment description, there are 500 years data are available and should use at least 200 years to obtain the statistically significant results. When presenting the climate change in Fig 6c-f, provide the statistical significance test.

Reference:

Roxy, M., Ritika, K., Terray, P. et al. Drying of Indian subcontinent by rapid Indian Ocean warming and a weakening land-sea thermal gradient. *Nat Commun* 6, 7423, 2015.

Burls, N. J., and Fedorov, A. V.: Wetter subtropics in a warmer world: Contrasting past and future hydrological cycles, *Proceedings of the National Academy of Sciences*, 114, 12888-12893, 10.1073/pnas.1703421114, 2017.

Reviewer #2:

Remarks to the Author:

Review of the manuscript entitled "Global warming-induced Asian hydrological climate transition across the Miocene-Pliocene boundary" by Ao et al.

Ao et al. produce a high-resolution geochemical record of Miocene to Pliocene from the Shilou section, CLP to investigate the link between global climate change and local environmental change. Using these newly acquired records, special analyses are carried out to examine the link between the orbital cycles and the variation of the paleoenvironment during this period. Climate models are also conducted to further support the inference of global warming drove climate change across the Miocene-Pliocene boundary.

Overall, the manuscript forces on an important question that would be of potential interest to the Asian paleoclimate community, I enjoy reading this manuscript. The text is well-written and figures are well-illustrated. But some of the interpretations and the limitations that should be improved and clarified before I can recommend the manuscript for publication in NC. I gave some of my concerns below for the authors' consideration.

My biggest concern is the way that authors tuning the age model. This is the fundamental of this paper. In addition, a few other studies on the almost same region established their age model based on magnetostratigraphy results without tuning their age model. Why you tuned the age model here? I am fine with tuning the age model if pessary although this is still challenged by many researchers. It seems that the authors only tuned 100-kyr cycles and did not tune obliquity and precession cycles. But why a further turning on obliquity and precession cycles should be over-tuned? The authors need to add more details about the rules of this tuning method. I think there might be a few correlation options when comparing the filtered 100-kyr eccentricity results with Laskar's data. I would recommend authors be more transparent by providing all these potential correlations (>30?) and the reason that choosing the one that presents in the text.

My second concern is that how to teeth out the tectonics impact from these geochemical records. In the current form of the manuscript, authors completely reject the possibility of tectonic uplift of Tibet and surrounding regions. Although briefly explain this inference in Line 373 to 388, it is not enough to support that tectonic is not relevant to this environmental change. If you look at the Low-T thermatology data in the surrounding region, there are many post-10 Ma ages showing the rapid exhumation in the surrounding region. Such as Clinkscales et al., 2020 *EPSL*; Su et al., 2021, Tectonics that focuses on the Shanxi rift, and many other papers that relate to the Qinling and Qilian Shan region. I would suggest the authors carefully examine the tectonic evolution of this region and fairly compiling all the ages that are related to tectonics.

Third, it seems that authors only present those records that are friendly to their model while that information that might contradict their model is ignored. For example, the authors cite Liu et al., 2014 *PNAS* geochemical record to support the spectral analysis however they ignore the evidence that clearly shows the intensified tectonic uplift in the Western Kunlun Shan, NW TP. A few recent studies

(Wang et al., 2019, Nature Communications; Wang et al., Science Advances.) that focus on the same region (Weihe Basin) do not show this environment shift at ~ 5.3 Ma. The authors failed to add these records into their regional proxies' correlation and even not cited these important findings. This is another clue that there might have some bias for the interpretation of these records in this study.

Fourth, I have no problem with the climate model itself and I do believe it is really a useful way that enables us to know what might happen in the past, especially examining some boundary conditions. However, for this manuscript, I do not think the climate simulation could add anything new to support the mechanism proposed to the study. By changing the boundary conditions, such as topography relief, CO₂ concentration, and other boundary conditions or even only change the spatial resolution, you will finally find a model that best-fits with what you want. Given the controversial topography relief in the surrounding mountain belts, precipitation, and many other factors, the interpretation should be mainly established from the paleoenvironmental records rather than from models.

Fifth, it is clear that this manuscript misses a statistical analysis section. As I mentioned in the previous comments, there could be a few uncertainties in the age model derived from the spectral analysis/tuning. Also, there are many unclear boundary conditions in the climate model. I would recommend the authors add a separate section describing all these uncertainties

Lastly, NC is one of the top-tier journals in Earth Science. Authors should be more transparent in publishing their data. I cannot find the raw data of these three records in the supplementary files. Reviewers and audience (once published) will be definitely like to evaluate the orbital cycles by running the spectral analysis themselves. I strongly suggest authors provide all the details of this work, including the raw data of the three records, details spectral analysis (the author said they have checked >30 tuning options in the methods section; what are they, show them in a Supplementary file), details of climate simulation.

Reply to Reviewer #1

Comment #1

P3. L45, also in P4 L60, About the Miocene-Pliocene boundary, when checking in the literature, the definition for the Miocene-Pliocene boundary has been mentioned frequently, but no literature defines the MPB at ~5.3 ma. If it is a new definition by this study, should clarify. Otherwise it would be misleading.

Response: Thank you for this helpful comment. The MPB definition used is not new to our study. We have clarified this in the revised manuscript with references (lines 57-60).

Comment #2

When presenting the data, the authors have had spectrum analysis and discussed the orbital variability in length. It is not very clear the role of orbital forcing in the MPB transition. My understanding is that the authors try to explain that the MPB transition is not due to the orbital forcing. If this is the case, considering most of the Nature Communication readers may not understand these forcing processes, this section can be largely shortened to avoid the confusion.

Response: Thank you for pointing this out. In addition to summer monsoon intensification across the MPB, prominent late Neogene orbital summer monsoon variability is a further important feature of our new palaeoclimate records. In the revised manuscript, we clarify that Asian climate change across the MPB is not due to orbital forcing (lines 223-225, 310-314). As suggested, we shortened this section to avoid confusion. Notably, the last paragraph in this section, which diverged to discuss implications of the 100-kyr monsoon variability over the last 1 Myr, has been removed.

Comment #3

P8 L73, What is "C3n.4n-C3r polarity reversal boundary"? It is hard to follow.

Response: To improve understandability, we have now changed "C3n.4n-C3r polarity reversal boundary" to "palaeomagnetic polarity reversal boundary between the C3n.4n normal polarity chron and the C3n.4r reversed polarity chron" (lines 165-166).

Comment #4

P23, L 275-285, The authors used oceanic proxy data in Indian Ocean to show the intensified summer monsoon, SSTs in Indian Ocean often affect the South Asian monsoon, when SST increase in these regions, it will decrease the land-sea contrast and weaken southerly wind, and weaken the South Asian monsoon (Roxy, et al., 2015), may provide less moisture to East Asia as well. I suggest the authors to discuss the changes in the land-sea contrast carefully.

Response: Thank you for this comment. Our simulations suggest that increased CO₂ enhanced land-sea contrast and westerly (20°N, 65-100°E) and southwesterly (20-35°N, 100-130°E) winds under Pliocene boundary conditions (Fig. 6e-f). Roxy et al. (2015) observed a decreased land-sea thermal gradient in South Asia in the context of current warming (different boundary conditions) because of rapid Indian Ocean warming and concurrent 'subdued' subcontinental warming. They produce this in

their simulations because they impose warming over the Indian Ocean region without warming the continent. In our Pliocene simulations, warming is larger over the continent than the ocean (Fig. 6c), so the mechanism of Roxy et al. (2015) does not apply in the situation that we are simulating.

Comment #5

In the climate modelling part, the only difference in the simulations for Miocene and Pliocene is CO₂ concentration (280 vs 400), this is similar to the present condition except the different boundary condition setting by PRISM4. Some discussion on the effect of the PRISM4 boundary condition is needed to clarify the climate response to CO₂ during MPB and present day.

Response: Thank you very much. We now discuss simulated CO₂-induced Asian climate changes under different boundary conditions for the MPB and present day in the “Methods” section of the revised manuscript (lines 492-507).

Comment #6

Burls and Fedorov (2017) used the same model CESM and did the simulation for Pliocene, they have drawn the opposite conclusion. Authors may compare with their work and discuss the possible reason why the same model simulate the opposite results.

Response: Burls and Fedorov (2017) used an atmospheric model forced with the sea surface temperature field from a fully coupled experiment with modified cloud radiative properties resembling early Pliocene reconstructions, but preindustrial topography and vegetation were imposed as boundary conditions in their Pliocene simulation. In contrast, we used a fully coupled model forced by reconstructed Pliocene orography, vegetation, ice sheets, lakes, and CO₂ concentration (400 ppm) from the PRISM4 dataset, which are more realistic. In addition to different models and boundary conditions, Burls and Fedorov (2017) used different model resolutions; they used T31 (~3.75°) atmospheric and land components coupled to ocean and ice components with a nominal 3° resolution, while we used ~2° atmospheric and land components coupled to ocean and ice components with a nominal 1° horizontal resolution. Finally, Burls and Fedorov (2017) did not investigate climate change when CO₂ concentration is increased in their Pliocene-like experiment. Their change between the Pliocene-like experiment and present day is opposite to that between an abrupt 4×CO₂ scenario and present day. We do not know what would have happened if they increased CO₂ using the Pliocene-like experiment as a baseline rather than the preindustrial. Thus, it is not really appropriate to compare our study with their work, and forcing such a comparison into the main text would confuse and distract from our core message.

For the sake of completeness, we assess spatial annual precipitation minus evaporation changes between preindustrial and Pliocene conditions in our simulations (see Figure 1 below), which is not included in the main text for reasons explained in the previous paragraph. We note that spatial annual precipitation minus evaporation changes between preindustrial and Pliocene conditions are different between our

simulations and those of Burls & Fedorov, which likely result from the aforementioned differences between models, model resolution, and boundary conditions.

Figure 1. Annual precipitation minus evaporation changes between our preindustrial and Pliocene simulations. Black dots denote regions statistically significantly above the 95% confidence level (Student's *t*-test).

Comment #7

The climate responses showed in Fig6 are based on the 30 years simulation data, to represent the climate change on million years time scale. Considering the well know multidecadal and centennial climate variability is present in both proxy data and model simulations, using 30 years mean is not enough. According to the model experiment description, there are 500 years data are available and should use at least 200 years to obtain the statistically significant results. When presenting the climate change in Fig 6c-f, provide the statistical significance test.

Response: Thank you. In the Pliocene Model Intercomparison Project (Haywood et al., 2011, 2016), most Pliocene simulations were run for 500 or >500 model years and the last 30, 50, or 100 model years were generally used to calculate climatological means (e.g., Haywood et al., 2011, 2016, 2020; Hunter et al., 2019; Chan and Abe-Ouchi, 2020; Feng et al., 2020; Samakinwa et al., 2020). For Pliocene simulations with 500-year integration lengths, the last ≥ 200 model years are less routinely used to calculate climatological means because the earlier part of the penultimate 100 model years may remain out of equilibrium. As is the case for routine Pliocene simulations, we reran the simulations and used the last 100 years to calculate statistically significant results instead of our previously used last 30 years. This also facilitates more precise comparison

between our and previous Pliocene simulations. We have clarified the model-run periods in lines 486–489. The comment about statistical significance is also much appreciated, and we have taken this opportunity to add statistical significance test results to Figure 6 and Supplementary Figure 11.

References cited in our response above

- Chan, W.L., Abe-Ouchi, A., 2020. Pliocene Model Intercomparison Project (PlioMIP2) simulations using the Model for Interdisciplinary Research on Climate (MIROC4m). *Climate of the Past* **16**, 1523–1545.
- Feng, R., Otto-Bliesner, B. L., Brady, E. C., Rosenbloom, N., 2020. Increased climate response and Earth system sensitivity from CCSM4 to CESM2 in mid-Pliocene simulations. *Journal of Advances in Modeling Earth Systems* **12**, doi: 10.1029/2019MS002033.
- Haywood, A.M., Dowsett, H.J., Robinson, M.M., Stoll, D.K., Dolan, A.M., Lunt, D.J., Otto-Bliesner, B., Chandler, M.A., 2011. Pliocene Model Intercomparison Project (PlioMIP): experimental design and boundary conditions (Experiment 2). *Geoscientific Model Development* **4**, 571–577.
- Haywood, A.M., Dowsett, H.J., Dolan, A.M., Rowley, D., Abe-Ouchi, A., Otto-Bliesner, B., Chandler, M.A., Hunter, S.J., Lunt, D.J., Pound, M., Salzmann, U., 2016. The Pliocene Model Intercomparison Project (PlioMIP) Phase 2: scientific objectives and experimental design. *Climate of the Past* **12**, 663–675.
- Haywood, A.M., Tindall, J.C., Dowsett, H.J., Dolan, A.M., Foley, K.M., Hunter, S.J., Hill, D.J., Chan, W.L., Abe-Ouchi, A., Stepanek, C., Lohmann, G., Chandan, D., Peltier, W.R., Tan, N., Contoux, C., Ramstein, G., Li, X.Y., Zhang, Z.S., Guo, C.C., Nisancioglu, K.H., Zhang, Q., Li, Q., Kamae, Y., Chandler, M.A., Sohl, L.E., Otto-Bliesner, B.L., Feng, R., Brady, E.C., von der Heydt, A.S., Baatsen, M.L.J., Lunt, D.J., 2020. The Pliocene Model Intercomparison Project Phase 2: large-scale climate features and climate sensitivity. *Climate of the Past* **16**, 2095–2123.
- Hunter, S.J., Haywood, A.M., Dolan, A.M., Tindall, J.C., 2019. The HadCM3 contribution to PlioMIP phase 2. *Climate of the Past* **15**, 1691–1713.
- Samakinwa, E., Stepanek, C., Lohmann, G., 2020. Sensitivity of mid-Pliocene climate to changes in orbital forcing and PlioMIP's boundary conditions. *Climate of the Past* **16**, 1643–1665.

Reply to Reviewer #2

Comment #1

My biggest concern is the way that authors tuning the age model. This is the fundamental of this paper. In addition, a few other studies on the almost same region established their age model based on magnetostratigraphy results without tuning their age model. Why you tuned the age model here? I am fine with tuning the age model if necessary although this is still challenged by many researchers. It seems that the authors only tuned 100-kyr cycles and did not tune obliquity and precession cycles. But why a further turning on obliquity and precession cycles should be over-tuned? The authors

need to add more details about the rules of this tuning method. I think there might be a few correlation options when comparing the filtered 100-kyr eccentricity results with Laskar's data. I would recommend authors be more transparent by providing all these potential correlations (>30?) and the reason that choosing the one that presents in the text.

Response: We are grateful for these helpful comments. We used an astronomical age model that was orbitally tuned to 100-kyr eccentricity from the magnetostratigraphy for the following reasons. First, the late Miocene–Middle Pliocene Rb/Sr record in the untuned magnetostratigraphy already shows prominent orbital variability (Supplementary Figs. 3 and 4), which warrants and facilitates orbital tuning. Furthermore, it is common practice to combine magneto- and astro-chronology to develop refined timescales for marine records (e.g., Coxall and Wilson, 2011; Drury et al., 2016; Holbourn et al., 2005; Lisiecki and Raymo, 2005; Pälike et al., 2001, 2006) and terrestrial records (e.g., Prokopenko et al., 2006; Sun et al., 2006; An et al., 2011). In addition, orbital expressions may be generally enhanced, and non-orbital signals subdued, in the refined astronomical age models relative to the untuned magnetostratigraphy, but the major orbital features are similar in both age models (Supplementary Fig. 4). This indicates that tuning was warranted and has not introduced spurious information – it merely served to provide chronological refinement and is common practice.

More specifically, the Rb/Sr record in the untuned magnetostratigraphy suggests a continuous 100-kyr eccentricity band throughout the late Miocene–Middle Pliocene, but only weaker and less continuous obliquity and precession bands (Supplementary Figs. 3 and 4). Accordingly, we only tuned the timescale using large-amplitude 100-kyr cycles (Supplementary Fig. 3), and refrained from tuning the obliquity and precession cycles to avoid over-tuning.

We have added more details on the tuning method and discuss associated uncertainties in the revised manuscript with numerous additional work, as suggested by Reviewer #2 (lines 412-457). We provide two other orbital tuning options with the youngest and oldest tuning age models (Supplementary Fig. 3; Supplementary Table 2). Our finally selected astronomical age model (intermediate age model) is balanced between the youngest and oldest tuning age models. The filtered 100-kyr Rb/Sr component correlates cycle-by-cycle with orbital eccentricity in all three tuning age models, with reasonable sedimentation rate changes, but ages of palaeomagnetic reversals are more consistent with their GPTS ages in the intermediate tuning age model (Supplementary Table 1), which was therefore selected as the final astronomical age model for the Shilou red clay sequence. For completeness, age uncertainties of the final astronomical age model are now calculated and presented in the revised manuscript (lines 447-449; Supplementary Fig. 3; Supplementary Table 2).

Comment #2

My second concern is that how to teeth out the tectonics impact from these geochemical records. In the current form of the manuscript, authors completely reject

the possibility of tectonic uplift of Tibet and surrounding regions. Although briefly explain this inference in Line 373 to 388, it is not enough to support that tectonic is not relevant to this environmental change. If you look at the Low-T thermometry data in the surrounding region, there are many post-10 Ma ages showing the rapid exhumation in the surrounding region. Such as Clinkscales et al., 2020 EPSL; Su et al., 2021, Tectonics that focuses on the Shanxi rift, and many other papers that relate to the Qinling and Qilian Shan region. I would suggest the authors carefully examine the tectonic evolution of this region and fairly compiling all the ages that are related to tectonics.

Response: Thank you for this comment. We now mention sedimentary records from the northern and eastern Tibetan Plateau (Fang et al., 2005; Fang et al., 2007; Métivier et al., 1998; Molnar, 2005), tectonic deformation in the Pamir and Tian Shan mountains (Fu et al., 2010; Hubert-Ferrari et al., 2007; Sun et al., 2009; Thompson et al., 2015; Wang et al., 2014), and low-temperature thermochronological data from the Shanxi Rift (North China), northern Tibetan Plateau, and Mongolian mountains (Clinkscales et al., 2020; De Grave et al., 2009; Liu et al., 2013; Peng et al., 2019; Su et al., 2021; Vassallo et al., 2007; Wang et al., 2011; Wang et al., 2020a). These show that late Miocene or Pliocene uplift of specific regions of the Tibetan Plateau and adjacent mountains occurred (lines 368-373). We no longer reject the possibility of tectonic uplift of Tibet and surrounding regions in the revised manuscript, but consider it as having a secondary role in Asian climate changes across the MPB (lines 373-374).

Comment #3

Third, it seems that authors only present those records that are friendly to their model while that information that might contradict their model is ignored. For example, the authors cite Liu et al., 2014 PNAS geochemical record to support the spectral analysis however they ignore the evidence that clearly shows the intensified tectonic uplift in the Western Kunlun Shan, NW TP. A few recent studies (Wang et al., 2019, Nature Communications; Wang et al., Science Advances.) that focus on the same region (Weihe Basin) do not show this environment shift at ~5.3 Ma. The authors failed to add these records into their regional proxies' correlation and even not cited these important findings. This is another clue that there might have some bias for the interpretation of these records in this study.

Response: We apologise for not mentioning these works in our original submission. These omissions have now been remedied; new references have been incorporated in the revised manuscript. We now discuss tectonic activity on the Tibetan Plateau and adjacent mountain ranges during the late Miocene–Pliocene, including tectonic evidence mentioned by Liu et al. (2014) and other studies (lines 368-373). We add a comparison between our work and existing studies that do not suggest summer monsoon intensification across the MPB (An et al., 2005; Clift et al., 2014; Wang et al., 2019; Wang et al., 2020b), including possible reasons for this difference (lines 263-276). The above additions do not contradict our major interpretations but make

our paper more comprehensive. We appreciate reviewer #2 for bringing these to our attention.

Comment #4

Fourth, I have no problem with the climate model itself and I do believe it is really a useful way that enables us to know what might happen in the past, especially examining some boundary conditions. However, for this manuscript, I do not think the climate simulation could add anything new to support the mechanism proposed to the study. By changing the boundary conditions, such as topography relief, CO₂ concentration, and other boundary conditions or even only change the spatial resolution, you will finally find a model that best-fits with what you want. Given the controversial topography relief in the surrounding mountain belts, precipitation, and many other factors, the interpretation should be mainly established from the paleoenvironmental records rather than from models.

Response: Despite potential uncertainties in recently reconstructed late Neogene boundary conditions, our simulations are crucial for grounding interpretations from proxy data. In this study, they provide important confirmation of our interpretations and especially of the physical consistency between records and potential forcing mechanism, as also highlighted by reviewer #1. Moreover, the broadly similar pattern of Asian climate response to CO₂-induced global warming between our late Neogene and previous preindustrial simulations suggests that the general pattern of Asian climate responses to CO₂ increase in our simulations may not change substantially even if the recently reconstructed late Neogene boundary conditions are moderately adjusted in future. Thus, we think that integrating proxy and modelling results strengthens our manuscript by establishing their consistency.

Comment #5

Fifth, it is clear that this manuscript misses a statistical analysis section. As I mentioned in the previous comments, there could be a few uncertainties in the age model derived from the spectral analysis/tuning. Also, there are many unclear boundary conditions in the climate model. I would recommend the authors add a separate section describing all these uncertainties.

Response: Thank you for suggesting that we include uncertainty analyses. We have added uncertainties in the age model derived from the spectral analysis/tuning in the revised manuscript (lines 447-449; Supplementary Fig. 3–4; Supplementary Tables 1–2). We also address uncertainties in late Neogene boundary conditions, simulation uncertainties due to boundary condition uncertainties, and potential influences on the major pattern of Asian climate responses to CO₂ increase in our present simulations if reconstructed late Neogene boundary conditions are adjusted slightly or moderately in future (lines 492-507). We took a month to run additional climate model simulations, to enable us to have a sound grip on uncertainties and significance.

Comment #6

Lastly, NC is one of the top-tier journals in Earth Science. Authors should be more transparent in publishing their data. I cannot find the raw data of these three records in the supplementary files. Reviewers and audience (once published) will be definitely like to evaluate the orbital cycles by running the spectral analysis themselves. I strongly suggest authors provide all the details of this work, including the raw data of the three records, details spectral analysis (the author said they have checked >30 tuning options in the methods section; what are they, show them in a Supplementary file), details of climate simulation.

Response: Thank you for this comment. We have added the raw data, orbitally tuned data, and details of orbital tuning and climate simulation in the “Methods”, Supplementary Information, and Supplementary Data.

References cited in our response above

- An, Z.S., Clemens, S.C., Shen, J., Qiang, X.K., Jin, Z.D., Sun, Y.B., Prell, W.L., Luo, J.J., Wang, S.M., Xu, H., Cai, Y.J., Zhou, W.J., Liu, X.D., Liu, W.G., Shi, Z.G., Yan, L.B., Xiao, X.Y., Chang, H., Wu, F., Ai, L., Lu, F.Y., 2011. Glacial-interglacial Indian summer monsoon dynamics. *Science* **333**, 719–723.
- An, Z.S., Huang, Y.S., Liu, W.G., Guo, Z.T., Clemens, S., Li, L., Prell, W., Ning, Y.F., Cai, Y.J., Zhou, W.J., Lin, B.H., Zhang, Q.L., Cao, Y.N., Qiang, X.K., Chang, H. and Wu, Z.K., 2005. Multiple expansions of C₄ plant biomass in East Asia since 7 Ma coupled with strengthened monsoon circulation. *Geology* **33**, 705–708.
- Clift, P.D., Wan, S.M. and Blusztajn, J., 2014. Reconstructing chemical weathering, physical erosion and monsoon intensity since 25 Ma in the northern South China Sea: a review of competing proxies. *Earth-Science Reviews* **130**, 86–102.
- Clinkscales, C., Kapp, P. and Wang, H.Q., 2020. Exhumation history of the north-central Shanxi Rift, North China, revealed by low-temperature thermochronology. *Earth and Planetary Science Letters* **536**, doi: 10.1016/j.epsl.2020.116146.
- Coxall, H.K., Wilson, P.A., 2011. Early Oligocene glaciation and productivity in the eastern equatorial Pacific: insights into global carbon cycling. *Paleoceanography* **26**, doi: 10.1029/2010PA002021.
- De Grave, J., Buslov, M.M., Van den Haute, P., Metcalf, J., Dehandschutter, B. and McWilliams, M.O., 2009. Multi-method chronometry of the Teletskoye graben and its basement, Siberian Altai Mountains: new insights on its thermo-tectonic evolution. In: *Thermochronological Methods: From Palaeotemperature Constraints to Landscape Evolution Models* (eds by Lisker, F., Ventura, B. & Glasmacher, U.A.) **324**, 237–259. Geological Society of London Special Publication.
- Drury, A.J., John, C.M., Shevenell, A.E., 2016. Evaluating climatic response to external radiative forcing during the late Miocene to early Pliocene: new perspectives from eastern equatorial Pacific (IODP U1338) and North Atlantic (ODP 982) locations. *Paleoceanography and Paleoclimatology* **31**, 167–184.
- Fang, X.M., Yan, M.D., Van der Voo, R., Rea, D.K., Song, C.H., Pares, J.M., Gao, J.P., Nie, J.S. and Dai, S., 2005. Late Cenozoic deformation and uplift of the NE Tibetan plateau: evidence from high-resolution magneto stratigraphy of the Guide Basin, Qinghai Province, China. *Geological Society of America Bulletin* **117**, 1208–1225.

- Fang, X.M., Zhang, W.L., Meng, Q.Q., Gao, J.P., Wang, X.M., King, J., Song, C.H., Dai, S. and Miao, Y.F., 2007. High-resolution magneto stratigraphy of the Neogene Huaitoutala section in the eastern Qaidam Basin on the NE Tibetan Plateau, Qinghai Province, China and its implication on tectonic uplift of the NE Tibetan Plateau. *Earth and Planetary Science Letters* **258**, 293–306.
- Fu, B.H., Ninomiya, Y. and Guo, J.M., 2010. Slip partitioning in the northeast Pamir-Tian Shan convergence zone. *Tectonophysics* **488**, 344–364.
- Holbourn, A., Kuhnt, W., Schulz, M., Erlenkeuser, H., 2005. Impacts of orbital forcing and atmospheric carbon dioxide on Miocene ice-sheet expansion. *Nature* **438**, 483–487.
- Hubert-Ferrari, A., Suppe, J., Gonzalez-Mieres, R. and Wang, X., 2007. Mechanisms of active folding of the landscape (southern Tian Shan, China). *Journal of Geophysical Research* **112**, doi: 10.1029/2006JB004362.
- Lisiecki, L.E., Raymo, M.E., 2005. A Pliocene–Pleistocene stack of 57 globally distributed benthic $\delta^{18}\text{O}$ records. *Paleoceanography* **20**, doi: 10.1029/2004PA001071.
- Liu, J.H., Zhang, P.Z., Lease, R.O., Zheng, D.W., Wan, J.L., Wang, W.T. and Zhang, H.P., 2013. Eocene onset and late Miocene acceleration of Cenozoic intracontinental extension in the North Qinling range-Weihe graben: insights from apatite fission track thermochronology. *Tectonophysics* **584**, 281–296.
- Métivier, F., Gaudemer, Y., Tapponnier, P. and Meyer, B., 1998. Northeastward growth of the Tibet plateau deduced from balanced reconstruction of two depositional areas: the Qaidam and Hexi Corridor basins, China. *Tectonics* **17**, 823–842.
- Molnar, P., 2005. Mio-Pliocene growth of the Tibetan Plateau and evolution of East Asian climate. *Palaeontologia Electronica* **8**, 1–23.
- Pälike, H., Norris, R.D., Herrle, J.O., Wilson, P.A., Coxall, H.K., Lear, C.H., Shackleton, N.J., Tripathi, A.K., Wade, B.S., 2006. The heartbeat of the Oligocene climate system. *Science* **314**, 1894–1898.
- Pälike, H., Shackleton, N.J., Rohl, U., 2001. Astronomical forcing in Late Eocene marine sediments. *Earth and Planetary Science Letters* **193**, 589–602.
- Peng, H., Wang, J.Q., Liu, C.Y., Zhang, S.H., Zattin, M., Wu, N. and Feng, Q., 2019. Thermochronological constraints on the Meso-Cenozoic tectonic evolution of the Haiyuan-Liupanshan region, northeastern Tibetan Plateau. *Journal of Asian Earth Sciences* **183**, doi: 10.1016/j.jseas.2019.103966.
- Prokopenko, A.A., Hinnov, L.A., Williams, D.F., Kuzmin, M.I., 2006. Orbital forcing of continental climate during the Pleistocene: a complete astronomically tuned climatic record from Lake Baikal, SE Siberia. *Quaternary Science Reviews* **25**, 3431–3457.
- Raymo, M.E., Lisiecki, L.E., Nisancioglu, K.H., 2006. Plio-pleistocene ice volume, Antarctic climate, and the global $\delta^{18}\text{O}$ record. *Science* **313**, 492–495.
- Su, P., He, H.L., Tan, X.B., Liu, Y.D., Shi, F. and Kirby, E., 2021. Initiation and evolution of the Shanxi Rift System in North China: evidence from low-temperature thermochronology in a plate reconstruction framework. *Tectonics* **40**, doi: 10.1029/2020TC006298.
- Sun, J.M., Li, Y., Zhang, Z.Q. and Fu, B.H., 2009. Magnetostratigraphic data on Neogene growth folding in the foreland basin of the southern Tianshan Mountains. *Geology* **37**,

1051–1054.

- Sun, Y.B., Clemens, S.C., An, Z.S., Yu, Z.W., 2006. Astronomical timescale and palaeoclimatic implication of stacked 3.6-Myr monsoon records from the Chinese Loess Plateau. *Quaternary Science Reviews* **25**, 33–48.
- Thompson, J.A., Burbank, D.W., Li, T., Chen, J. and Bookhagen, B., 2015. Late Miocene northward propagation of the northeast Pamir thrust system, northwest China. *Tectonics* **34**, 510–534.
- Vassallo, R., Jolivet, M., Ritz, J.F., Braucher, R., Larroque, C., Sue, C., Todbileg, M. and Javkhlanbold, D., 2007. Uplift age and rates of the Gurvan Bogd system (Gobi-Altay) by apatite fission track analysis. *Earth and Planetary Science Letters*, **259**, 333–346.
- Wang, G.C., Cao, K., Zhang, K.X., Wang, A., Liu, C., Meng, Y.N. and Xu, Y.D., 2011. Spatio-temporal framework of tectonic uplift stages of the Tibetan Plateau in Cenozoic. *Science in China* **54**, 29–44.
- Wang, H.L., Lu, H.Y., Zhao, L., Zhang, H.Y., Lei, F. and Wang, Y.C., 2019. Asian monsoon rainfall variation during the Pliocene forced by global temperature change. *Nature Communications*, **10**, doi: 10.1038/s41467-019-13338-4.
- Wang, W.T., Zheng, D.W., Li, C.P., Wang, Y., Zhang, Z.Q., Pang, J.Z., Wang, Y., Yu, J.X., Wang, Y.Z., Zheng, W.J., Zhang, H.P. and Zhang, P.Z., 2020a. Cenozoic exhumation of the Qilian Shan in the northeastern Tibetan Plateau: evidence from low-temperature thermochronology. *Tectonics* **39**, doi: 10.1029/2019TC005705.
- Wang, X., Sun, D.H., Chen, F.H., Wang, F., Li, B.F., Popov, S.V., Wu, S., Zhang, Y.B. and Li, Z.J., 2014. Cenozoic paleo-environmental evolution of the Pamir-Tien Shan convergence zone. *Journal of Asian Earth Sciences* **80**, 84–100.
- Wang, Y.C., Lu, H.Y., Wang, K.X., Wang, Y., Li, Y.X., Clemens, S., Lv, H.Z., Huang, Z.H., Wang, H.L., Hu, X.Z., Lu, F.Z. and Zhang, H.Z., 2020b. Combined high- and low-latitude forcing of East Asian monsoon precipitation variability in the Pliocene warm period. *Science Advances* **6**, doi: 10.1126/sciadv.abc2414.

Reviewers' Comments:

Reviewer #1:

Remarks to the Author:

The authors have made effort to re-do the analysis and revise the manuscript according to my comments, I am happy with the clarification and revision. The manuscript is now presented in a clear way and can consider to be published. I am not an expert on the clay sediment, therefore I leave the evaluation on the process and interpretation of the paleoclimate data to the reviewer who has the expertise.

For the final publication, a thorough language check may need, for example, present tense and past tense are mixed sometimes, should be consistent. Some oral presentation may adjust to scientific writing, for example, use simulations instead of runs.

Reply to Reviewer #1

Comment #1

For the final publication, a thorough language check may need, for example, present tense and past tense are mixed sometimes, should be consistent. Some oral presentation may adjust to scientific writing, for example, use simulations instead of runs.

Response: We have performed further detailed language checking. Use of past tenses is used in relation to geological events and the present tense is used in relation to modern conditions. Usages have been checked and modified to ensure accuracy and consistency. All colloquial wordings have been replaced with more formal wordings.